# OpenReview forum: "ChatInject: Abusing Chat Templates for Prompt Injection in LLM Agents"
_ICLR.cc/2026/Conference — ICLR 2026 Poster_

### Official Review · Reviewer_FxRy · 2025-10-28

**Soundness:** 3
**Presentation:** 3
**Contribution:** 4
**Rating:** 6
**Confidence:** 3

**Summary:**

The paper proposes ChatInject, a family of indirect prompt-injection attacks against LLM agents that exploit chat-template role tags (e.g., system/user/assistant delimiters) and persuasive multi-turn dialogues to bypass instruction hierarchies. Compared to plain-text injections, ChatInject wraps attacker content to mimic native chat templates; a multi-turn variant ("Multi-turn + ChatInject") primes the model across turns. Across InjecAgent and AgentDojo benchmarks, the authors report large gains in attack success rate (ASR)  and show that attacks transfer across models, including closed-source ones, and evade common prompt-based defenses.

**Strengths:**

+ The paper frames how forged role tags inside tool outputs can be misinterpreted as higher-priority instructions, bypassing system, user, assistant hierarchies—distinct from prior plain-text injections.

+ The four variants—Default InjecPrompt, InjecPrompt+ChatInject, Default Multi-turn, and Multi-turn+ChatInject—are defined with a consistent template.

+ The transferable attack is an interesting and important angle, which is worth further study.

**Weaknesses:**

- While behaviorally persuasive, the paper lacks a deeper mechanistic explanation of why template tokens grant authority (e.g., attention patterns, representation shifts).

- Multi-turn dialogues are LLM-generated (GPT-4.1) and human-reviewed, but not drawn from real attacker corpora. This leaves ecological validity open: would organic, messy content sustain similar gains?

**Questions:**

1. If the same role tags are injected but tokenized differently (e.g., via Unicode homoglyphs or byte-level encodings), how do models react?

2. Have you validated multi-turn persuasion with human-written or naturally occurring sequences (e.g., scraped forum/HTML excerpts) to assess ASR/Utility differences vs. GPT-generated dialogues?

3. Table 2 shows higher transfer to certain closed models but less Utility degradation in some settings. Can you hypothesize which agent-stack choices (e.g., tool arbitration, post-filters) helped preserve Utility? Any evidence from ablations?

---

> ### Author Response · Authors · 2025-11-20
> **Response to Reviewer FxRy (1 / 2)**
>
> Your detailed comments are highly appreciated. We have reviewed and incorporated all your feedback into the revised paper.
>
> ---
>
> ### Response to Weakness 1 - Detailed Explanation of Chat Template Authority
> > While behaviorally persuasive, the paper lacks a deeper mechanistic explanation of why template tokens grant authority (e.g., attention patterns, representation shifts).
>
> Building on our findings about the relationship between embedding similarity and ASR, we additionally conducted attention analysis to understand the mechanistic basis of why template tokens grant authority. Following prior work [1] showing that role indicators can reshape attention patterns, we measured how chat-template tags redistribute attention between the user instruction and the attacker instruction.
>
> We use the conversation logs between the real user and the assistant from InjecAgent benchmark, and annotate each utterance with an explicit role tag so that the input more closely reflects how the LLM perceives the content. Using the last input token as the query, we then compute attention over all previous tokens and extract only the portions directed toward user vs. attacker instructions. Their relative weighting is: attn(user_instruction) / (attn(user_instruction)+attn(attacker_instruction)).
>
> | Model | Attack type | User Instruction | Attacker Instruction |
> |:-:|:-:|:-:|:-:|
> | Qwen-3 | InjecPrompt | 52.62 | 47.38 |
> | Qwen-3 | ChatInject | 45.54 | 54.46 |
> | GPT-oss | InjecPrompt | 51.41 | 48.59 |
> | GPT-oss | ChatInject | 32.10 | 67.90 |
>
> Across models, **ChatInject consistently shifts attention toward attacker instructions.** This attention reallocation explains why template-formatted malicious content achieves higher priority—the model fundamentally changes how it allocates computational resources when processing template-wrapped payloads. These results are well aligned with the experimental findings reported in [1]. We provided the results in **Appendix C.3** in the revised version.
>
> ---
>
> ### Response to Weakness 2 & Questions 2 - Human-like Multi-turn Injection
> > Multi-turn dialogues are LLM-generated (GPT-4.1) and human-reviewed, but not drawn from real attacker corpora. This leaves ecological validity open: would organic, messy content sustain similar gains?
>
> > Have you validated multi-turn persuasion with human-written or naturally occurring sequences (e.g., scraped forum/HTML excerpts) to assess ASR/Utility differences vs. GPT-generated dialogues?
>
> Human-generated multi-turn datasets exist only for jailbreak attacks, not for prompt injection. As in **Response to Weakness 1** for **Reviewer hF66**, the structural and objective differences between these two attack scenarios mean existing jailbreak datasets cannot be directly applied to our task. To address this limitation, we adapted the multi-turn jailbreak dataset from Li et al. [2] by using GPT-4.1-based modification to reframe attack objectives for the prompt injection setting—specifically, transforming dialogues to guide agents toward executing attacker instructions via tool calls rather than eliciting harmful content. All adapted dialogues were manually reviewed to ensure they fit our task requirements. We then compared this adapted real attacker corpora against our persuasion-based multi-turn approach on AgentDojo, reporting Attack Success Rate (ASR):
>
> |                         | qwen3 | glm-4.5 |
> |-------------------------|:-------:|:---------:|
> | Persuasion (ours)       | 80.5  | 48.1    |
> | Real attacker corpora   | 76.3  | 27.5    |
>
> **Our persuasion-based approach achieves superior ASR**, demonstrating that our synthetic generation methodology produces effective and realistic multi-turn attacks. We provided the results in **Appendix C.1** in the revised version.
>
> ---
>
> [1] Zhang et al., On the loss of context-awareness in general instruction fine-tuning.
>
> [2] Li et al., LLM Defenses Are Not Robust to Multi-Turn Human Jailbreaks Yet

---

> ### Author Response · Authors · 2025-11-20
> **Response to Reviewer FxRy (2 / 2)**
>
> ### Response to Questions 1 - Different Tokenization Methods
> > If the same role tags are injected but tokenized differently (e.g., via Unicode homoglyphs or byte-level encodings), how do models react?
>
> In **Section 5.1**, we show that the more similar the injected chat template is to the model’s own chat template, the more effectively it can mislead the LLM, leading to a higher ASR. Intuitively, if a change in tokenization still allows the model to recognize the semantics of the template so that the embedding similarity remains high, we would expect the ASR to stay high as well; and vice versa. To examine whether our conclusions extend to alternative tokenization schemes, we conducted an additional experiment in which we encoded the templates using **Unicode homoglyphs**.
>
> | Model | Similarity | ChatInject (Homoglyphs)  | ChatInject |
> |-|:-:|:-:|:-:|
> | Qwen-3 | 0.326 | 17.5 | 54.8 |
> | GPT-oss | 0.468 | 0.3 | 51.4 |
> | Llama-4 | 0.657 | 1.5 | 17.2 |
>
> From these results, we observe that the homoglyph-encoded chat templates **have almost no effect on the models, yielding very low ASR values**. This can be explained by their low similarity to the original templates: for all three models, the **homoglyph variants exhibit even lower similarity scores than the least similar “foreign” templates reported in Figure 3**, and their ASR scores are correspondingly the lowest. Based on this, we infer that for other tokenization methods as well, attacks will remain effective only when the embedding similarity to the original template is preserved; if the similarity substantially decreases, the resulting attack becomes much less effective. We added results and the following analysis in the **Appendix C.4** in the revised version.
>
> ---
>
> ### Response to Questions 3 - Relatively Small Utility Degradation on Closed-LLMs
> > Table 2 shows higher transfer to certain closed models but less Utility degradation in some settings. Can you hypothesize which agent-stack choices (e.g., tool arbitration, post-filters) helped preserve Utility? Any evidence from ablations?
>
> Closed-source models are only accessible via API, limiting our ability to analyze their internal mechanisms. We cannot determine what agent-stack components, prompt engineering beyond the user-accessible system prompt, or post-processing filters they employ. This makes it impossible to identify specific architectural choices that preserve utility.
> However, for models like Claude, where the system prompt is publicly documented [3], we tested whether this component alone contributes to Utility preservation. We applied Claude’s released system prompt to open-source models (Qwen3-235B-A22B-2507 and GLM-4.5) and compared Utility under ChatInject to their Default InjecPrompt configurations:
>
> | Attack                 | qwen3-235b-a22b-2507 | glm-4.5 |
> |------------------------|:-----------------------:|:---------:|
> | Default                | 50.9                  | 78.4    |
> | ChatInject             | 28.3 (-22.6)          | 67.9 (-10.5) |
> | + claude system prompt | 32.4 (-18.5)          | 65.8 (-12.6) |
>
> **The Claude-style system prompt produced only marginally smaller Utility drops**, and the pattern was inconsistent across models. Beyond this minor effect, we did not observe any additional mechanisms that could explain the Utility preservation seen in some closed models. We added results and the following analysis in the **Appendix C.5** in the revised version.
>
> ---
>
> Once again, we sincerely appreciate your detailed comments on our paper. Please share any further concerns with us.
>
> [3] https://docs.claude.com/en/release-notes/system-prompts

---

### Official Review · Reviewer_rfLo · 2025-10-31

**Soundness:** 4
**Presentation:** 4
**Contribution:** 3
**Rating:** 8
**Confidence:** 4

**Summary:**

The paper proposes a novel type of indirect prompt injection attacks against language models (LMs) deployed in agentic settings. The attack exploits the prompt templates used by the models. More specifically, the proposed prompt injections are wrapped as single- or multi-turn conversations in the LM native prompt template using the special tokens from the template. This results in a "privilege escalation," where text that is supposed to be treated by the LM as a tool output gets interpreted as a higher-level user or system instruction.

The attack is demonstrated to be highly effective against several near-frontier LMs on two benchmarks. Injections also transfer non-trivially to LMs with different prompt templates. A method for making the attack universal against an unknown template is also demonstrated. Several existing defenses are tried against the new attack. Finally, a separate defense of detecting the malicious template in the payload is considered, and a counter-attack with randomizing injection characters is evaluated.

**Strengths:**

- The paper is clearly written, with well-defined contributions and a logical red team-blue team structure. The figures are also helpful and aesthetically pleasing.
- The attack vector is, to my knowledge, novel.
- Evaluation is comprehensive (6 open-source LLMs and 3 closed-source LLMs). Reasonable ablations are performed (single- vs multi-turn, tool call hooks, reasoning hooks).
- I appreciated the fine-grained analysis of ASR against prompt template similarity, which clearly shows that the attack success is driven by guessing the correct template, rather than some unknown mechanism related to OOD inputs.
- A method for making the attack independent of the template shows that even an attacker with a few attempts before being discovered and blocked by the provider might succeed with this attack.
- Reasonable blue team counteractions are considered, although I am less certain here. The counter-measure of detecting prompt templates  and counter-counter-measure of introducing noise are very logical.

**Weaknesses:**

I'm not sure if the discussion of the possible defenses in Section 6.1 is comprehensive. For example, the paper discusses the ProtectAI prompt injection detector and shows that, although ASR under this detector drops significantly, high FPR of the detector results in significant utility drops. Since the paper tests closed-source LMs for vulnerabilities, it might also be worth it to try proprietary defense mechanisms against prompt injections, like the Lakera Guard.

Generally, the paper does not always report error bars, and when it does, the bars seem rather wide, especially given that std is reported instead of 95% CI. For the ASR values in tables 1 & 2 and in Fig. 3, one simple way to report the CI would be to use the Wilson interval formula, which only depends on the ASR and the size N of the dataset. Given that rerunning the queries might be expensive, this could provide a cheap proxy for at least one source of the noise (due to dataset size but not due to LLM generation distribution). I understand that the tables could then become cluttered, so the confidence intervals for at least some of the values could be put in the appendix. Relatedly, one expects the pi detector to result in consistently lower ASR. This is true for all bars but one (multi-turn with Grok 3). Is this outlier due to noise? Having the CI would help here too.

Current related work discussion is too small. If the paper is accepted, I would recommend to spend some of the extra page allowance on expanding to include more work on indirect prompt injections, instruction hierarchy, and other relevant topics, eg [1,2,3]. Prompt injection literature is rather rich.

[1] Kai Greshake, Sahar Abdelnabi, Shailesh Mishra, Christoph Endres, Thorsten Holz, and Mario Fritz.
Not what you’ve signed up for: Compromising real-world llm-integrated applications with indirect
prompt injection. In Proceedings of the 16th ACM workshop on artificial intelligence and security,
pp. 79–90, 2023.

[2] Zverev, Egor, et al. "Can llms separate instructions from data? and what do we even mean by that?." arXiv preprint arXiv:2403.06833 (2024).

[3] Florian Tramer, Nicholas Carlini, Wieland Brendel, and Aleksander Madry. On adaptive attacks to
adversarial example defenses. Advances in neural information processing systems, 33:1633–1645,
2020.


Minor nit: L374 "remains strong transferability" -> "retains strong transferability"

**Questions:**

- What are the data sample sizes for the two benchmarks that the paper considers? This could be useful to report in Sec. 3.3 (this subsection would also fit better in the beginning of Section 4 in my opinion, although this is a minor point).
- What is the reason for missing values in Table 1? My guess is that this is is due to the models not having the capabilities to think / perform tool calls. It would be good to report in the table caption.

---

> ### Author Response · Authors · 2025-11-20
> **Response to Reviewer rfLo (1 / 2)**
>
> Thank you for the positive review of our paper. We are also very grateful for your deep interest and constructive advance warnings. We have clearly addressed all your points and incorporated them into this revision.
>
> ---
>
> ### Response to Weakness 1 - Limited Defense Methods
> > I'm not sure if the discussion of the possible defenses in Section 6.1 is comprehensive. For example, the paper discusses the ProtectAI prompt injection detector and shows that, although ASR under this detector drops significantly, high FPR of the detector results in significant utility drops. Since the paper tests closed-source LMs for vulnerabilities, it might also be worth it to try proprietary defense mechanisms against prompt injections, like the Lakera Guard.
>
> To more comprehensively evaluate how to defend against ChatInject-style attacks, we additionally conducted experiments using one of the proprietary defense mechanisms you suggested, namely **Lakera Guard**. In doing so, we designed two strategies for handling prompt injection and evaluated both. The first strategy, **Filtering**, follows the same paradigm as our original pi detector–based defense: we replace the external detector from Protect AI with Lakera Guard and filter out any tool outputs flagged as malicious. The second strategy, **Masking**, instead removes only those segments within the tool output that are identified as malicious, using the full conversation between the original user and the agent system as context.
>
> We applied these two Lakera-Guard–based defense strategies to Qwen-3 and Grok-3, and recorded ASR and Utility for each attack type:
>
> | Model | Strategy  | Metric | InjecPrompt | ChatInject | Multi-turn | Multiturn + ChatInject |
> |--------|-----------|---------|:-----------:|:----------:|:----------:|:-----------------------:|
> | Qwen-3 | Filtering | ASR | 0.5 | 1.5 | 1.8 | 2.3 |
> | Qwen-3 | Filtering | Utility | 17.5 | 18.8 | 20.1 | 19.0 |
> | Qwen-3 | Masking | ASR | 24.9 | 51.6 | 37.5 | 76.1 |
> | Qwen-3 | Masking | Utility | 50.1 | 48.1 | 36.0 | 34.7 |
> | Grok-3 | Filtering | ASR | 0.3 | 0.3 | 0.3 | 0.3 |
> | Grok-3 | Filtering | Utility | 22.6 | 21.6 | 22.1 | 20.9 |
> | Grok-3 | Masking | ASR | 10.3 | 34.4 | 31.6 | 54.2 |
> | Grok-3 | Masking | Utility | 74.0 | 58.1 | 69.9 | 65.8 |
>
> From these results, we first observe that the **Filtering** strategy substantially reduces ASR, but **at the cost of a large drop in Utility**. This mirrors our findings with the original pi detector and illustrates a fundamental limitation of filtering-based defenses: once content is flagged as malicious even a single time, the entire tool output is removed, effectively stalling the agentic pipeline. As a result, Utility degradation is difficult to avoid, even when the detector’s false positive rate is relatively low. We will add the filtering-based defense results with Lakera Guard to **Figure 5** and include a corresponding analysis in **Section 6.1**.
>
> By contrast, the **Masking** strategy mitigates the drop in Utility to some extent, but **its reduction in ASR is also much weaker**. We attribute this to the fact that Lakera Guard’s fine-grained detection remains imperfect, so it often fails to precisely isolate and remove only the malicious segments, thereby allowing attacks to succeed.

---

> > ### Comment · Reviewer_rfLo · 2025-11-26
> > **Response to authors**
> >
> > I appreciate the detailed response and the additional experiments on Lakera guard.
> >
> > > To more comprehensively evaluate how to defend against ChatInject-style attacks, we additionally conducted experiments using one of the proprietary defense mechanisms you suggested, namely Lakera Guard.
> >
> > Great to see this introduced. Looks like the guard does successfully mitigate the attacks, but also, similarly to pi-detector, reduces utility. On further reflection, I'm not as sure if utility, as described in L205-206 is a very useful metric. If an agent that I run is under attack, I would prefer it to shut down immediately and report the incident to me, rather than continue operating. I did not notice this subtlety on the previous read and thought that utility is measured under benign instructions _without attack_. Another framework that has to deal with a trade-off between safety under adversarial attacks and utility is the AI control framework [1] (the attacker here is an AI rather than human, but some similarities do exist), and it is common there, and to my taste more natural, to define utility as performance when no attack is present. I do agree, however, that there still might be some circumstances, where good performance when under attack is necessary, e.g. if in the near future prompt injections become so omnipresent on the internet that any agent interacting with the Web is forced to deal with them at all times.
> >
> > I see now how this distinction is responsible for the drop in utility under injection detectors, and how this is not necessarily related to a high FPR.
> >
> > I think this part of the definition should be highlighted in the text and discussed more explicitly, or other readers might come away as confused as I was.
> >
> > [1] Greenblatt, Ryan, et al. "AI control: Improving safety despite intentional subversion." arXiv preprint arXiv:2312.06942 (2023).
> >
> > > For all remaining experiments, the LLM temperature is fixed to 0, making the outputs deterministic and obviating the need for variance-based reporting.
> >
> > I believe that this is incorrect; the "hidden" source of randomness in most evaluations is the dataset itself, not only the LLM. By evaluating on a dataset of a fixed size, you substitute true expected value of some random variable by the empirical mean, which is why techniques like bootstrapping exist.
> >
> > > In Figure 4, the error bars for the single-template setting appear relatively wide because, when the injected template coincides with the target model’s own template, the ASR spikes and becomes much higher than in the other cases
> >
> > This is insightful, I think it would be nice to highlight this more in discussion in Section 5.3.
> >
> > > Following your suggestion, we additionally computed 95% confidence intervals using the Wilson interval for our main experiments and incorporated them into Figure 3 and the Appendix D.5. Finally, the relative robustness of the Default Multi-turn injection to the pi detector is due to limitations of the detector in handling persuasive dialogue payloads rather than noise in the data, as noted in line 475~477. We once again appreciate your feedback, which helped us improve the clarity and rigor of our analysis.
> >
> > Thank you, this makes the results in Figure 3 look more trustworthy.
> >
> > > We have expanded the related work section as suggested.
> >
> > I appreciate that, the discussion now looks more comprehensive.
> >
> > Overall, these issues do not feel to me like they are sufficient to lower the score.

---

> > > ### Comment · Reviewer_rfLo · 2025-11-26
> > > **Weaknesses highlighted by other reviewers**
> > >
> > > I'd also like to flag that I did study the weaknesses highlighted by the reviewers who gave a negative score to this paper and the authors' responses, and I do not find them to be persuasive or warrant a rejection. To discuss some of them,
> > >
> > > > Contribution is only in the attack side, nothing in the defense side.
> > >
> > > The authors, in my opinion correctly, perform 1 or 3 steps in the monitor-attacker game, so this is misguided. The attacker moves second [2]. Furthermore, contributions on the attack side only are legitimate and sufficient, if executed well.
> > >
> > > [2] Nasr, Milad, et al. "The attacker moves second: Stronger adaptive attacks bypass defenses against llm jailbreaks and prompt injections." arXiv preprint arXiv:2510.09023 (2025).
> > >
> > > > The main limitations are lack of novelty and lack of baselines
> > >
> > > I tend to agree with the authors's response here that the proposed baselines are not about prompt injections.
> > >
> > > > The technical novelty is limited: the work mainly enhances IPI by reformatting or rephrasing injected content, and the ideas of leveraging chat templates or persuasive multi-turn dialogues have been conceptually explored in prior works.
> > >
> > > One expects to see the references in a comment like this, especially if it is used as grounds for a rating of 2.
> > >
> > > > ChatInject and Multi-turn components appear somewhat loosely connected —one manipulates structure, the other context; they are combined mainly for empirical gain without a unified theoretical justification.
> > >
> > > I'm not sure if a "theoretical justification" is strictly necessary (as in, one should reject a paper without it) if the attacks work.

---

> > > ### Author Response · Authors · 2025-12-01
> > > **Response to follow-up question by Reviewer rfLo**
> > >
> > > We thank you for your continued engagement and the positive feedback on our revision. We are glad that the additional experiments with Lakera Guard and the expanded related work section were well-received.
> > >
> > > ### Response to Concern 1 - Utility Under Attack & Detector-based Defense
> > > We would like to clarify that our paper includes utility measurements under benign (non-attack) conditions, which can be found in **Table 4** under the “Benign Utility” column. As you noted, the rationale for measuring utility under attack—consistent with prior work—is to assess whether an agent can robustly complete the user’s intended task even when malicious payloads are present. This captures an important dimension: resilience to attacks without completely shutting down functionality. We appreciate your reference to the AI control framework and agree that both perspectives on utility have merit depending on the threat model. In response to your comment, we have added a short discussion in **lines 931–941** that explicitly clarifies the role of each metric in our evaluation.
> > >
> > > ### Response to Concern 2 - Randomness of the Dataset
> > > We initially treated the dataset used in our experiments as a static component and assumed that greedy decoding would yield fully deterministic outputs. However, as you correctly pointed out, regardless of determinism at the model level, the benchmark dataset itself can be viewed as a finite sample from the underlying instruction distribution. To explicitly reflect the uncertainty arising from this dataset sampling, we have added 95% confidence intervals computed using the Wilson interval in **Appendix E** (While re-running all experiments with full dataset bootstrapping to obtain exact uncertainty estimates would be computationally expensive, we followed your suggestion and used the Wilson interval to capture the overall uncertainty in a principled yet efficient manner).
> > >
> > > ### Response to Concern 3 - Wide Error Bar in Figure 4
> > > We are very pleased to hear that our analysis provided useful insight. As you suggested, we highlighted this analysis in **Section 5.3**.
> > >
> > > ---
> > >
> > > We believe these additional clarifications and discussions have further strengthened the paper. We are genuinely grateful for your constructive feedback throughout this review process.

---

> ### Author Response · Authors · 2025-11-20
> **Response to Reviewer rfLo (2 / 2)**
>
> ### Response to Weakness 2 - Measuring Variability
> > Generally, the paper does not always report error bars, and when it does, the bars seem rather wide, especially given that std is reported instead of 95% CI. For the ASR values in tables 1 & 2 and in Fig. 3, one simple way to report the CI would be to use the Wilson interval formula, which only depends on the ASR and the size N of the dataset. Given that rerunning the queries might be expensive, this could provide a cheap proxy for at least one source of the noise (due to dataset size but not due to LLM generation distribution). I understand that the tables could then become cluttered, so the confidence intervals for at least some of the values could be put in the appendix. Relatedly, one expects the pi detector to result in consistently lower ASR. This is true for all bars but one (multi-turn with Grok 3). Is this outlier due to noise? Having the CI would help here too.
>
> We thank the reviewer for the insightful comments on the statistical analysis of our results. **The difference between the settings where we report standard deviation and those where we do not is whether the outcome is stochastic.** In **Figures 4 and 6**, randomness is involved (the ordering of model templates in **Figure 4** and which characters are perturbed within the template in **Figure 6**), so we report standard deviation to capture the stability of these results.
>
> In **Figure 4**, the error bars for the single-template setting appear relatively wide because, when the injected template coincides with the target model’s own template, the **ASR spikes and becomes much higher than in the other cases**. In **Figure 6**, the standard deviation reflects how ASR fluctuates depending on which characters are perturbed, but even at the lower end of this range the ASR remains above that of the no-template baseline, so the effectiveness of perturbation is preserved. For all remaining experiments, the LLM temperature is fixed to 0, making the outputs deterministic and obviating the need for variance-based reporting.
>
> Following your suggestion, we additionally computed 95% confidence intervals using the Wilson interval for our main experiments and incorporated them into **Figure 3**,  **Figure 5**, and the **Appendix E**. Finally, the relative robustness of the Default Multi-turn injection to the pi detector is due to **limitations of the detector in handling persuasive dialogue payloads** rather than noise in the data, as noted in **line 475~477**. We once again appreciate your feedback, which helped us improve the clarity and rigor of our analysis.
>
> ---
>
> ### Response to Weakness 3 - Small Related Works
> > Current related work discussion is too small. If the paper is accepted, I would recommend to spend some of the extra page allowance on expanding to include more work on indirect prompt injections, instruction hierarchy, and other relevant topics, eg [1,2,3]. Prompt injection literature is rather rich.
>
> We have expanded the related work section as suggested. In the revision, we incorporated the recommended references and enriched the discussion on indirect prompt injections, instruction hierarchy, and multi-turn attacks.
>
> ---
>
> ### Response to Questions 1 - Data Sample Sizes
> > What are the data sample sizes for the two benchmarks that the paper considers? This could be useful to report in Sec. 3.3 (this subsection would also fit better in the beginning of Section 4 in my opinion, although this is a minor point).
>
> For the InjectAgent benchmark, we use 17 user instructions, 30 direct-harm attacker instructions, and 32 data-stealing attacker instructions. We apply all pairwise combinations of user and attacker instructions to the agent system, resulting in 17 × (30 + 32) = 1,054 samples in total. For the AgentDojo benchmark, we use 21 user instructions and 5 attacker instructions in the Slack domain, 20 user instructions and 7 attacker instructions in the Travel domain, and 16 user instructions and 9 attacker instructions in the Banking domain. This yields 21 × 5 + 20 × 7 + 16 × 9 = 389 user–attacker pairs in total. We have updated **Section 3.3** to explicitly report these benchmark statistics.
>
> ---
>
> ### Response to Questions 2 - Reason for Missing Values in Table 1
> > What is the reason for missing values in Table 1? My guess is that this is is due to the models not having the capabilities to think / perform tool calls. It would be good to report in the table caption.
>
> We thank the reviewer for this helpful observation. As you correctly pointed out, in Table 1 we evaluate the reasoning hook and the tool-calling hook only on models that explicitly provide <think> template tokens and <tool_call> template tokens, respectively. We have added this clarification to the caption of **Table 1**.
>
> ---
>
> We once again thank you for your constructive suggestions, which have helped us refine and further advance our work. We welcome any remaining concerns you might have.

---

> ### Author Response · Authors · 2025-11-30
>
> We deeply appreciate your careful examination of the concerns raised by other reviewers and your fair assessment of our responses. Your support regarding the novelty and scope of our work is truly valued.

---

### Official Review · Reviewer_5Lup · 2025-11-01

**Soundness:** 2
**Presentation:** 3
**Contribution:** 2
**Rating:** 2
**Confidence:** 4

**Summary:**

This paper presents ChatInject, a new attack that enhances indirect prompt injection (IPI) in LLM-based agents by exploiting chat template structures (e.g., \<system\>, \<user\>, \<assistant\> tags) and multi-turn persuasive dialogues.
The first component (ChatInject) formats malicious payloads to mimic the model’s native chat template, making the model misinterpret the injected content as higher-priority instructions.
The second component (Multi-turn variant) uses persuasion-driven multi-turn dialogues to gradually legitimize malicious requests.
Extensive experiments on InjecAgent and AgentDojo benchmarks across 9 LLMs (both open- and closed-source) show substantial improvements in attack success rate (ASR) and cross-model transferability. The paper also evaluates standard defenses, finding them largely ineffective.

**Strengths:**

- The paper is clear and well-organized.
- The proposed attack is effective, increasing the ASRs significantly across different datasets and backbone models.
- Comprehensive experiments across multiple models, datasets, and defenses provide strong empirical support. The analysis is rich, covering template similarity, cross-model transfer, and robustness to perturbations, making the study thorough from an empirical perspective.

**Weaknesses:**

- The technical novelty is limited: the work mainly enhances IPI by reformatting or rephrasing injected content, and the ideas of leveraging chat templates or persuasive multi-turn dialogues have been conceptually explored in prior works.
- ChatInject and Multi-turn components appear somewhat loosely connected —one manipulates structure, the other context; they are combined mainly for empirical gain without a unified theoretical justification.

**Questions:**

- Could a simple defense that detects the presence of special chat template tokens in external content effectively identify such attacks without significantly reducing utility?

---

> ### Author Response · Authors · 2025-11-20
> **Response to Reviewer 5Lup (1 / 2)**
>
> We are thankful for your interest in our study and for the several issues you have highlighted, especially in theoretical aspects. We will address the areas identified as weaknesses with detailed explanations and theoretical justification, ensuring a comprehensive response.
>
> ---
>
> ### Response to Weakness 1 - Lack of Novelty
> > The technical novelty is limited: the work mainly enhances IPI by reformatting or rephrasing injected content, and the ideas of leveraging chat templates or persuasive multi-turn dialogues have been conceptually explored in prior works.
>
> As specific references were not provided in the review regarding the 'prior works,' we address this concern by differentiating our work from the most relevant existing studies.
>
> **Chat Template Exploitation:** While prior work such as ChatBug [1] (as detailed in line 107) also uses chat templates, **both the core mechanism and the target task are fundamentally different.** ChatBug demonstrates that replacing a model's native chat template with foreign template can break safety alignment in jailbreak scenarios. In contrast, our work shows that exploiting the model's native chat template—specifically, the instruction-following behavior learned during training—causes the model to obey malicious instructions embedded in tool outputs. Moreover, our work addresses indirect prompt injection, not jailbreak, making it a distinct contribution in both approach and application domain.
>
> **Multi-turn in Indirect Prompt Injection:** Multi-turn attacks are well-established in jailbreak [2], but they have never been effective in indirect prompt injection. When we attempt to apply existing plain-text–based methods to multi-turn settings, they provide no benefit—as shown in **Table 1** (Default Multi-turn column), where ASR remains at 0.1% for GPT-oss and GLM-4.5 on InjecAgent. In contrast to these ineffective plain-text approaches, our method wraps each conversational turn in role-specific tags, which enables multi-turn attacks to achieve 65.9% ASR on Qwen-3 and 88.3% on Llama-4 (Table 1, Multi-turn + ChatInject). To the best of our knowledge, this is the **first work to successfully enable multi-turn attacks in indirect prompt injection through chat template exploitation.**
>
> We respectfully request that the reviewer provide specific citations to support the claim that these ideas have been "conceptually explored in prior works," as this would enable us to better position our contributions and address any overlooked related work.
>
> ---
>
> ### Response to Weakness 2 - Connection Between ChatInject and Multi-turn
> > ChatInject and Multi-turn components appear somewhat loosely connected —one manipulates structure, the other context; they are combined mainly for empirical gain without a unified theoretical justification.
>
> The ChatInject and Multi-turn components are tightly coupled through a **causal relationship: chat template structure is the necessary prerequisite that enables multi-turn contextual manipulation in prompt injection settings.**
>
> In standard prompt injection, agents process tool outputs as a single, low-priority token sequence. When multi-turn dialogues are embedded as plain text within this sequence, the model perceives them as passive content rather than active conversational turns. This occurs because the dialogue structure is flattened into a mere string, preventing the model from recognizing distinct "User" and "Assistant" interaction history.
>
> ChatInject addresses this by injecting specific role tokens that artificially reconstruct the turn-taking mechanism. Since LLMs are instruction-tuned to rely on these tags for identifying dialogue structure and segmenting turns, **this structural manipulation effectively exploits the model's learned dependency.** It forces the model to interpret the injected text not as tool-returned data, but as a valid, authoritative conversational history with proper role hierarchies. Only through this template-based segmentation can the contextual persuasion mechanisms of **multi-turn dialogues become active.** As a result, this combination raises the average ASR of plain-text multi-turn approaches on both the InjecAgent and AgentDojo benchmarks from 13.8% to 45.6%, as in Table 1. We have incorporated this a priori justification of the connection between ChatInject and multi-turn attacks into the Introduction.
>
> ---
>
> [1] Jiang et al., Chatbug: A common vulnerability of aligned llms induced by chat templates.
>
> [2] Weng et al., Foot-in-the-door: A multi-turn jailbreak for llms.

---

> ### Author Response · Authors · 2025-11-30
> **Response to Reviewer 5Lup (2 / 2)**
>
> ### Response to Questions 1 - Defense by Format Stripping
> > Could a simple defense that detects the presence of special chat template tokens in external content effectively identify such attacks without significantly reducing utility?
>
> As you correctly anticipated, ChatInject can indeed be defended against relatively easily using a rule-based template-stripping mechanism (counter), as we already acknowledge in the paper **(line 503~506)**. To address this limitation, we conduct the template perturbation (counter-counter) experiments in **Section 6.2**, where we intentionally modify the template format in an unusual way. As stated in **line 507~510**, once the template has been perturbed, it becomes impossible to rule-basedly detect the original chat template of any given model. For example, original Qwen system tag “<|im_start|>system” becomes “<|isart|sysm” after removal perturbation. In **Figure 6**, we further show that such **perturbed chat templates still yield a higher ASR** than the setting without any template. Consequently, for perturbed variants of ChatInject, simple detection of special chat template tokens is insufficient to identify this attack type.
>
> ---
>
> Thank you once again for your thorough feedback. We believe that we have addressed your main concerns about the paper. Please let us know if anything remains unresolved. We have also revised the **Introduction** to reflect your comments, and we would greatly appreciate it if you could take a look at the updated version.

---

### Official Review · Reviewer_hF66 · 2025-11-01

**Soundness:** 2
**Presentation:** 2
**Contribution:** 2
**Rating:** 4
**Confidence:** 3

**Summary:**

This paper introduces ChatInject, a novel indirect prompt injection attack that exploits chat template structures used in large language model (LLM) agents. Unlike prior prompt injection work focusing on plain-text manipulation, ChatInject forges system/user/assistant role tags to bypass instruction hierarchies and embeds malicious commands in persuasive multi-turn dialogues. Through extensive experiments on nine LLMs (both open- and closed-source) across AgentDojo and InjecAgent benchmarks, the authors show that ChatInject achieves dramatically higher attack success rates and maintains strong transferability even without template knowledge. They also show existing defenses are largely ineffective, motivating the need for stronger mitigations.

**Strengths:**

- Novel threat model with multi-turns persuasive which is indeed a solid area of study and contribution.
- Comprehensive evaluation and Strong empirical results
- Clear writing and strong figures: Figures and tables effectively demonstrate the mechanisms and results (e.g., Figs. 1–6).

**Weaknesses:**

- The main limitations are lack of novelty and lack of baselines. - The contributions in prompt creation seems incremental, limiting its novelty. For example a missing related work: xteaming: https://arxiv.org/pdf/2504.13203 or Foot-In-The-Door, which is referred in the paper but not considered in the exp.

- The treat model needs more clarity: I did not undergard from which point the attacker gets access to change the prompts.

- Contribution is only in the attack side, nothing in the defense side.
- I am not if the utility gets lower then how they are not detected easily? Intuitively, this should be easier to detect.
- Table 2 is hard to follow, i think adding a simple avg per row can make it more readable.

- Synthetic dialogue generation: Multi-turn persuasion data are LLM-generated (GPT-4.1) and manually filtered; realism and diversity might

- Interpretability gap: In Fig 1 I found the prompts with not readbale tokens but reading the contribution does not give clear idea when they are not readbale and when. Alos this makes it less interpretable and somewhat detectable.

**Questions:**

See weaknesses

---

> ### Author Response · Authors · 2025-11-20
> **Response to Reviewer hF66 (1 / 2)**
>
> Thank you for taking a deep interest in our paper and for pointing out potential issues in advance. We have revisited the points you raised and the direction of our paper, and we would like to address them clearly.
>
> ---
> ### Response to Weakness 1 - Clarifying Task Distinctions and Contributions
> > The main limitations are lack of novelty and lack of baselines. - The contributions in prompt creation seems incremental, limiting its novelty. For example a missing related work: xteaming: https://arxiv.org/pdf/2504.13203 or Foot-In-The-Door, which is referred in the paper but not considered in the exp.
>
> The two **referenced works**, x-teaming and Foot-In-The-Door, **address multi-turn jailbreaking scenarios, which fundamentally differ from our focus on indirect prompt injection.** **Jailbreaking** relies on **direct, interactive adversarial user inputs** where the attacker actively drives the conversation flow to bypass safety alignment. In contrast, **indirect prompt injection** occurs via **passive external tool outputs**, where the attacker cannot interactively engage with the agent turn-by-turn. Due to this structural distinction, strictly **applying interactive jailbreak baselines** to a non-interactive injection setting is structurally **infeasible**.
>
> To demonstrate this fundamental incompatibility, we applied existing plain-text prompt injection methods to multi-turn settings. As shown in **Table 1** under the "Default Multi-turn" column, these approaches **remain largely ineffective** (e.g., 0.1% ASR on GPT-oss and GLM-4.5 on InjecAgent). This result confirms that simply adopting **multi-turn dialogue** structures from **jailbreak literature does not transfer** to prompt injection without addressing a critical challenge.
>
> Our work addresses this challenge through **two interconnected contributions**. First, we exploit chat template structures by **mimicking role tags that bypass instruction hierarchies**. Second, we enable effective multi-turn prompt injection by **embedding simulated multi-turn dialogues within a single injection payload** using these chat templates, overcoming the limitation of passive tool outputs. This approach achieves substantially higher ASRs on InjecAgent, improving from an average of 7.72% to 52.33% across models.
>
> Furthermore, regarding the Foot-In-The-Door baseline, we provide **additional analysis in Appendix C.1, Figure 7b**, where we evaluate our approach across different persuasion taxonomies, demonstrating robustness across diverse multi-turn strategies.
>
> ---
>
> ### Response to Weakness 2 - Threat model clarity
> > The treat model needs more clarity: I did not undergard from which point the attacker gets access to change the prompts.
>
> Following the standard threat models established in InjecAgent and AgentDojo, **the attacker can only access and modify tool outputs.** This represents the most minimal and realistic assumption for indirect prompt injection attacks, where adversaries inject malicious content through external data sources (e.g., API responses) that the agent retrieves via tool calls. We have added explicit clarification of this threat model in **Section 3.1** to make this assumption clear from the outset.
>
> ---
>
> ### Response to Weakness 3 - Contribution in Defense Side
> > Contribution is only in the attack side, nothing in the defense side.
>
> As shown in **Figure 5**, we find that existing self-resilient defenses are ineffective against this type of attack, effectively forcing practitioners to rely on external detectors that risk degrading task utility. Also, as in **Section 6.2**, considering a straightforward **counter-defense (Format Stripping)** in which the defender is aware of our attack, we show that an attacker can respond with **counter-counter-attack (Adversarial Perturbation)**, successfully evading such rule-based chat template detection while still achieving a non-trivial ASR. Taken together, these analyses highlight that future defense research should move toward template-aware methods that detect and handle injected payloads without harming utility.

---

> ### Author Response · Authors · 2025-11-20
> **Response to Reviewer hF66 (2 / 2)**
>
> ### Response to Weakness 4 - Relationship Between Utility Drop and Detection
> > I am not if the utility gets lower then how they are not detected easily? Intuitively, this should be easier to detect.
>
> Consistent with prior prompt injection work [1,2], post-hoc detection is not considered meaningful in our setting. **As noted in line 206,** a drop in utility is observed only after all necessary tools have been executed and the entire agent session has concluded; by this point, the damage is irreversible (e.g., private data has already been sent). Therefore, even when utility decreases, it does not constitute effective detectability within our threat model.
>
> ---
>
> ### Response to Weakness 5 - Clarity of Table 2
> > Table 2 is hard to follow, i think adding a simple avg per row can make it more readable.
>
> We agree that providing overall average ASR values for each model and each template would make the table clearer. We have added an average row and an average column for each model and template in **Table 2** and updated the accompanying discussion.
>
> ---
>
> ### Response to Weakness 6 - Synthetic Dialogue Generation
> > Synthetic dialogue generation: Multi-turn persuasion data are LLM-generated (GPT-4.1) and manually filtered; realism and diversity might
>
> Regarding diversity, we evaluated our approach across **four distinct persuasion techniques** as shown in **Appendix C.1, Figure 7(b)**. Our results demonstrate that ChatInject consistently achieves high ASR across all persuasion strategies.
>
> Regarding realism, our work is the first to introduce simulated multi-turn dialogues for indirect prompt injection attacks, meaning no existing human-generated corpora exist for this specific setting. Therefore, to validate the realism of our synthetic approach, we adapted the **human-written multi-turn jailbreak dataset** from Li et al., [3] by using GPT-4.1-based modification to reframe attack objectives for the prompt injection setting—specifically, transforming dialogues to guide agents toward executing attacker instructions via tool calls. All adapted dialogues were manually reviewed to ensure they fit our task requirements. We then compared this adapted real attacker corpora against our persuasion-based multi-turn approach on AgentDojo, reporting ASR:
> |                         | qwen3 | glm-4.5 |
> |-------------------------|-------|---------|
> | Persuasion (ours)       | 80.5  | 48.1    |
> | Real attacker corpora   | 76.3  | 27.5    |
>
> **Our persuasion-based approach achieves superior ASR**, demonstrating that our synthetic generation methodology produces effective and realistic multi-turn attacks. We added results and the following analysis in the **Appendix C.5** in the revised version.
>
> ---
>
> ### Response to Weakness 7 - On the Use of Non-Readable Tokens
> > Interpretability gap: In Fig 1 I found the prompts with not readbale tokens but reading the contribution does not give clear idea when they are not readbale and when. Alos this makes it less interpretable and somewhat detectable.
>
> **Interpretability gap:** We would like to clarify that there are no inherently "unreadable tokens" in our approach. We believe the reviewer may be referring to the chat template special tokens visible in **Figure 1** (e.g., <|im_start|>, <|im_end|>). These are not unreadable artifacts but rather predefined, model-specific tokens that LLMs are explicitly trained on to parse conversational structure and role hierarchies. These template tokens are essential to our attack mechanism because they enable the model to segment and interpret injected content according to learned role priorities (system > user > assistant > tool output).
> If the reviewer is referring to a different aspect of **Figure 1** or has concerns about specific elements that appear unclear, we would greatly appreciate further clarification.
>
> **Regarding detectability:** The use of predefined template formats may raise the possibility that simple rule-based filters could detect the attack. To evaluate this concern, **Section 6.2** examines whether ChatInject remains effective when template patterns are corrupted to evade detection. We applied 10% character-level perturbations to the templates, and as shown in **Figure 6**, the attack still consistently outperforms the baseline Default InjecPrompt across all tested models. This demonstrates that ChatInject remains an effective attack even when obfuscated to avoid simple pattern-based detection.
>
> ---
>
> Once again, we sincerely appreciate your detailed comments on our paper. Please inform us of any remaining concerns you might have. We incorporated your detailed reviews in the revised version.
>
> [1] Zhang et al., Agent Security Bench (ASB): Formalizing and Benchmarking Attacks and Defenses in LLM-based Agents.
>
> [2] Debenedetti et al., AgentDojo: A Dynamic Environment to Evaluate Prompt Injection Attacks and Defenses for LLM Agents.
>
> [3] Li et al., LLM Defenses Are Not Robust to Multi-Turn Human Jailbreaks Yet.

---

### Author Response · Authors · 2025-11-26
**Summary of Our Work and Discussions**

We thank the reviewers and Area Chair. We summarize key points and revisions below.

---

### Structural Necessity of ChatInject (Reviewer 5Lup)
We clarify that ChatInject acts as a structural prerequisite for multi-turn attacks. Since models are instruction-tuned on specific formats, they rely on a learned dependency on role tokens to identify dialogue structure. ChatInject activates this dependency by artificially reconstructing the turn-taking hierarchy within tool outputs. This forces the model to interpret the payload as authoritative history rather than passive data, thereby enabling multi-turn persuasion. Only through this template-based segmentation can multi-turn persuasion become effective. We have revised the **Introduction and Related Work** sections to explicitly clarify the threat model.

### Clarifying Misunderstandings
- **Distinction from Standard Jailbreaking (Reviewer hF66, 5Lup):** Our work fundamentally differs from prior jailbreak methods. We clarified that unlike these interactive jailbreaking methods, Indirect Prompt Injection (IPI) occurs via passive tool outputs where the attacker cannot actively drive the conversation. As a result, both the template usage and the conclusions we draw are qualitatively different from template-based jailbreak attacks (e.g., ChatBug), and existing multi-turn jailbreak baselines (e.g., Foot-in-the-Door) alone fail to substantially increase ASR in the IPI setting (**Table 1**). To the best of our knowledge, no prior work has combined these ideas in the prompt injection setting to improve attack efficiency, and the reviewers also did not provide prior works for this task.
- **Unreadable Tokens (Reviewer hF66):** We clarify that Figure 1 contains no unreadable tokens. The reviewer may be referring to the chat template role tags, which are fully readable by both humans and LLMs. These are predefined, model-specific tokens that LLMs are explicitly trained to parse for conversational structure. They are not ignored by the model but processed together with the payload.
- **Defense Side (Reviewer hF66, 5Lup):** Several comments on the defense side stemmed from overlooking analyses that are already present in the paper. As discussed in **Section 6.2**, we already instantiate one- and three-step variants of the monitor–attacker game, a point that **Reviewer rfLo** explicitly acknowledged. We consider the seemingly straightforward counter-defense of rule-based template stripping and show that a counter-counter strategy via adversarial template perturbation can evade such detection while still achieving non-trivial ASR (**Figure 6**).
- **Regarding Utility (Reviewer hF66, rfLo):** We report both benign utility (**Table 4**) and utility under attack, following prior work, to capture whether an agent can still complete the user’s intended task in the presence of malicious payloads instead of simply “shutting down” under strict defenses. We argue that a drop in utility is not a viable detection metric because the damage (e.g., data exfiltration) often occurs before the task concludes. Therefore, post-hoc detection based on utility loss is ineffective. Additional details regarding this rationale are provided in **Appendix C.7**.


### Additional Materials
- **Minor Complements (Reviewer hF66, rfLo):** We have clarified the reasons for the missing values in **Table 1** directly in the caption, and we have added average rows and columns to **Table 2** to improve readability. Finally, we explicitly highlight in **Section 5.3** why the error bars in **Figure 4** appear relatively wide.

- **Human-like Multi-turn Payload (Reviewer hF66, FxRy):** We adapted a human-curated multi-turn jailbreak dataset and conducted comparative experiments against our synthetic dialogues. The detailed setup and results are provided in **Appendix C.1**.

- **Additional Defense Method (Reviewer rfLo):** We conducted further experiments using Lakera Guard. The detailed setup and results are reported in **Section 6.1** and **Figure 5**.

- **Dataset Size and Confidence Intervals (Reviewer rfLo):** We have added a detailed breakdown of dataset sizes by domain in **Section 3**. Furthermore, to statistically capture the uncertainty arising from dataset sampling, we compute 95% confidence intervals using the Wilson interval based on the sample counts, and report them in **Figure 3**, **Figure 5**, and **Appendix E**.

- **Attention Analysis (Reviewer FxRy):** We performed an attention distribution analysis. The detailed setup and results are presented in **Appendix C.3**.

- **Different Tokenization (Reviewer FxRy):** We applied homoglyph encoding to the chat templates and re-ran the prompt-injection experiments. The detailed setup and results are given in **Appendix C.4**.

- **Utility Degradation in Closed-LLMs (Reviewer FxRy):** We transferred Claude’s system prompt into an open-source LLM and evaluated the resulting behavior. The detailed setup and results are provided in **Appendix C.5**.

---

### Meta-Review · Area_Chair_t8zX · 2026-01-06

**Summary:**

This paper presented a study to abuse chat template / role tokens for more effective prompt injection. There are several main concerns for this paper:

1. Multiple reviewers are concerned about the limited novelty of the paper, as it primarily reformat the injected contents
2. Concern about the synthetic multi-turn data not being realistic
3. Concern about not having enough experiments / discussions on defenses
4. Question regarding the relation between multi-turn and ChatInject
5. Question about deeper mechanistic explanation

**Reviewer Concerns:**

1. The authors provided clarification about the novelty. While it is still debatable how much novelty this paper brings (esp. technical novelty) in the context of previous prompt injection work, this paper does provide an interesting perspective towards this problem.
2. The authors provided additional information regarding the diversity and realism of the generate multi-turn dataset.
3. The authors clarified a few results of non-effectivenss of baseline defense
4. The author clarified the causal dependency
5. The authors provided additional analysis

**Reviewer Scores:**

I expect Reviewer hF66 to slightly increase the score as their concerns are mostly clarified, especially with the analysis on the diversity and realism of the synthetic dataset.

Reviewer 5Lup may slightly increase their score. Their question on the relation between multi-turn and ChatInject is mostly clarified. But depending on how they believe about the novelty of the paper, they may or may not increase the score beyond the acceptance threshold.

Reviwer rfLo and Reviewer FxRy will likely keep their positive score.

---

### Decision · Program_Chairs · 2026-01-26

Accept (Poster)